# Understanding of Public–Private Partnership Stakeholders as a Condition of Sustainable Development

**Anna Wojewnik-Filipkowska** [1,*]  **and Joanna Węgrzyn** [2]

1   Department of Investment and Real Estate, Faculty of Management, University of Gdańsk,
    Armii Krajowej 101, 81-824 Sopot, Poland, anna.filipkowska@ug.edu.pl
2   Department of Economics of Real Estate and Investment, Cracow University of Economics,
    Rakowicka 27a, 31-510 Cracow, Poland; wegrzynj@uek.krakow.pl
*   Correspondence: anna.filipkowska@ug.edu.pl; Tel.: +48-58-523-11-74

**Abstract:** The strategic goal of city management is to ensure its sustainable development which requires a balance of rare resources. From the operational perspective, namely projects implementing sustainable development, the balance refers to human resources. They can be classified into the public or private sector and their cooperation is known as Public–Private Partnership (PPP). Building on the concept of sustainable development and stakeholder theory, the research develops a conceptual framework of stakeholder analysis in PPP projects. More generally, the research aims to contribute to a theoretical understanding of the determinants of sustainable city development and PPP success factors. The research claims that the PPP procurement is consistent with sustainable urban development and the PPP model, accompanied by the stakeholder theory, requires evaluation which balances diverse stakeholders' interests along the triple bottom of sustainable development. The conceptual framework combines stakeholder attributes of preferred benefits and power and urgency. It includes a time and scope perspective. The research has a descriptive but also a normative character as the framework could be helpful to understand and engage stakeholders in sustainable urban development. The developed framework can be considered for the future construction of a model that can be implemented and tested. This theoretical research is based on a literature survey, applying methods of critical analysis and construction. The innovative approach of the research is based on integrated application of already known concepts of sustainable development, stakeholder theory, and Public–Private Partnership, which are all necessary to create a new approach to management of city development consistent with the known facts.

**Keywords:** Public–Private Partnership; infrastructure projects; sustainable development; stakeholder analysis; human resource; social capital; project evaluation

## 1. Introduction

The concentration of population, economic activity, social, technical, administrative and political issues relate to cities and therefore the future of humankind is linked with cities. The cities thus have to be managed for their inhabitants to ensure their sustainable development in the condition of rare resources. Sustainable development is demonstrated by qualitative changes in the structure of the economy, availability of goods and services for citizens, leading to a better standard of living and an increase in national income. It requires simultaneous consideration of the three pillars (triple bottom) [1] namely the economy, society, and the environment, regardless of the specific objectives of each city or country. A balanced life model concerning economic growth is connected with an

even distribution of benefits. A long-term and responsible growth is to be shared by all entities, communities, and nations. Economic goals are then expressed by a stable national economy, satisfying basic needs by providing sustainable products at appropriate prices, counteracting concentration and economic power, internalizing external costs, and minimizing the import of raw materials. Socially sustainable development means access to goods and services concerning social diversity, without violating cultural wealth, so that every society can shape its own future [2]. Social objectives also apply to civil society and participatory democracy [3] which are fundamental to cooperation. In the area of protection of natural resources and the environment for future generations, it is necessary to search and implement solutions which reduce resource consumption, limit environmental contamination, and protect natural ecosystems.

This research focuses on Public–Private Partnership (PPP), in the context of sustainable development as a strategic goal of city development, and underlines the balanced use, creation, and maintenance of scarce human resources at the operational level. Human and social resources are a distinct kind of resources as, according to endogenous growth theories, participation and engagement of different stakeholders determine sustainable development [4] and the resources will become capital only when they generate certain benefits [5]. The participation and engagement can be enabled, and benefits can be created via cooperation between the public and private sector, known as PPP projects. The sustainable infrastructure projects execute a strategy of sustainable city development on the operational level [6].

With a background in sustainable development, research is driven by the necessity of a **new approach to stakeholder analysis in PPP projects** highlighting the need to understand and engage PPP stakeholders in sustainable urban development. The research assumes that the understanding and engagement in a project by a single stakeholder influences the project's success and therefore sustainable development. We also assume that stakeholder analysis requires a new approach because of the characteristics of PPP projects. Finally, referring to Kauko, Siniak, and Źróbek [7] we assume that sustainable investment management is more important than government regulations on sustainability as a sustainable investment is a tool for sustainable city development.

Figure 1 presents the research questions (RQ), the aim, and the structure.

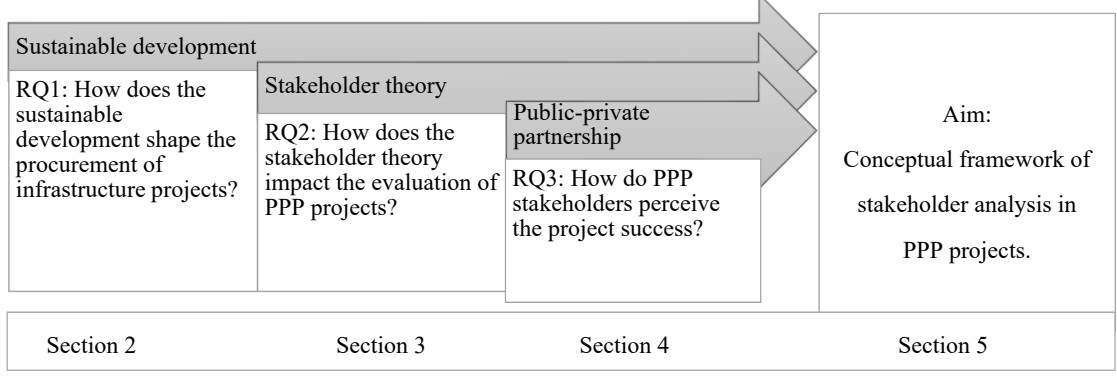

**Figure 1.** Research questions, aim, and structure (source: own study).

More generally, research aims to contribute to a theoretical understanding concerning the determinants of sustainable city development and success of PPPs. We identified the research gap relating stakeholder analysis in PPP projects in the sustainability context. There is a limited number of research works concerning interdisciplinary debates about the concepts of sustainable urban development [7,8]. The issue of stakeholders engagement in PPP projects is also a rare subject of research [9–12]. Finally, Prandecki et al. [13] claim that urbanization, public networking, and institutional changes (all related to the subject of the research) will have the most significant impact on sustainable development. Bjärstig [14] said that only a few authors of the collaborative governance literature have focused on sustainability outcomes and also according to Hueskes et al. [1] there is a

need to include sustainability in PPP projects. Miccoli, Finucci, and Murro [15] also call for integration of the development process, which means among others a holistic approach, bottom-up planning and implementation, community, and Public–Private sector financial collaboration. We would like to contribute to this discussion [16–19] as well as shed some more light on the reasons for PPP success as well as failure. Although the National Audit Office [20] states that a dramatic drop in both the number and the capital value of new PPP projects in the United Kingdom has been determined by the financial crisis and the increased costs of private finance, the complex relationships in PPP have been reported as one of the main reasons for a PPP failure [21–24]. Among others, Schepper et al. [11] claim that there is a gap between PPP stakeholders expectations and the outcome of the project which determines the project success, and Guarini and Battisti [10] state that processes implemented through PPP need balance between public and collective interests. The novelty of this research relies more on a new application of current state-of-art than on inventing new ideas from scratch as the general idea behind this research was to integrate different concepts together to make a breakthrough in PPP stakeholders understanding and therefore PPP project success and implementation in sustainable urban development.

Given the above background and justification, we used sustainable development as an analytical starting point in this paper. It is organized in sections corresponding with the research questions and the research aim. The introductory section is followed by the literature review related to sustainable development in Section 2 and the specifics of PPP in the context of stakeholder theory in Section 3. Section 4 refers to PPP stakeholders' perception of success. Finally, the interactions between sustainable development, stakeholder theory, and Public–Private Partnership enabled us to build and discuss the conceptual framework of stakeholder analysis in PPP projects. The article closes with conclusions in Section 6. The conceptual framework might be a recommendation for policymakers, urban developers, public administration, and citizens to strengthen collaboration supporting city sustainable development. The proposed framework may be viewed as a theoretical reference for the future methodological and operational applications relating to infrastructure projects. The research has thus a descriptive but also a normative character. It is based on a literature review, including "classics" on sustainable development (Brundtland (WCED) [25], Strange and Bayley (OECD) [26]) and social capital (Coleman [27]), stakeholder theory (Freeman [28,29], Savage et al. [30], Brenner [31], Mitchel, Agle, and Wood [32]), and Public–Private Partnership (Bulljevich and Park [33], Flyvbjerg et al. [17]) as well as newer authors, presenting some of the dilemmas relating to multidimensional success evaluation [34–36]. The literature review includes also local authors publishing locally in their native language [2,4,5,37,38], reports [20,39,40], and guides [41,42] to enrich the research. The research applies methods of critical analysis and construction.

## 2. Resource Context of Sustainable Development Shaping Procurement of Infrastructure Projects

Sustainable development "meets the needs of the present without compromising the ability of future generations to meet their own needs" [25] (p. 41). The original focus of the term on the "needs" has been recently replaced by the concept of the "rights" and sustainable development can be interpreted regarding "integrated" orders pursuing economic growth, protecting the natural capital, and promoting social justice [43] (p. 4). The second focus of the sustainable development concept refers to limitations and abilities related to the needs of current and future generations. These limitations and abilities involve integrated and **rare resources.** Sustainable use, construction, and maintenance of these are determined by the level of technological development and social organization.

The issue of resources in sustainable development is raised by Strange and Bayley [26]. They explain the necessity of planning and managing economic, human and natural resources to improve current societies without worsening the situation for future generations. They call attention to cooperation as the well-being of economies, people and environment are linked. This connection means that the economic, social, and environmental aspects are interrelated and must be considered together [14]. In particular, neither economic development alone is enough to solve social problems, nor

can environmental problems be solved only by technical means [44]. Secondly, the connection means the necessity of managing the development over geographic, institutional, and sectoral boundaries which is a challenge.

Wojewnik-Filipkowska [6], inspired among others by Girard [8,45] and his multidimensional approach to city management, distinguishes four dimensions of sustainable development, including a resource point of view. First, sustainability relates to the aforementioned balance of the three pillars of sustainable development (economic growth, social justice, and protection of the environment). Second, in the context of city development, the balance relates to different types of infrastructure (economic, business, social, and environmental protection). Third, it relates to the balance between "hard" components and "soft" values as the city is both a physical place of paths and buildings, but also a space of values, beliefs, and relations. Finally, sustainable development refers to a balance of scarce resources. These embrace produced and economic resources including financial ones, the creative capital of the city including human and social resources, and natural resources including space within the cities.

Concerning resources, Matysiak [5] points out the superiority of financial resources in the current economy as it is money that provides access to all other resources. However, this situation is changing, because knowledge, which is an attribute of human resources, and relations, which are attributes of social capital, are becoming increasingly important. Human resources refer to total individual predisposition of each person, which includes motivation, intellectual predispositions, and competences. Motivation is the result of a combination of acquired knowledge, willingness to work, and essential skills. Intellectual predisposition is the ability to think creatively, imagination, and the ability to react quickly. Competencies are theoretical knowledge and experience. Human resources are then indicated by education, work experience, tenure, and on-the-job training. They are contained in networks and used in human activities [46] and social interactions, which can be interpreted as a form of social capital [27]. Trust, credibility, and moral norms, which are related to human resource and social capital, materialize and determine cooperation.

Finally, concerning human resources and social capital in the sustainable development context, Dixon claims: "Instead of striving for physical growth, a city's success today should be measured by how wisely it uses energy, water, and other resources, how well it maintains a high quality of life for its people, and how smart it is in building prosperity on a sustainable foundation" [47] (p. 2). It is people (human resource) with their knowledge (attribute of social resource) who are responsible for improving the usage of all resources including themselves, creating human and social capital. Sustainable development should then focus on wise and improved usage of the resources [7,26] which can be interpreted as a smart sustainable development [6,48] confirming synergies of the sustainability [19,49,50]. "Smart" relates (among others) to people (social and human resources) and governance (participation) [45]. Smart people means that the city residents, "who are the greatest value of the city" [51] (p. 361), initiate and implement the changes by their engagement and participation. Smart people and smart governance are therefore components of a sustainable development, shaping the procurement of infrastructure projects towards Public–Private cooperation, instead of a conventional form of public sector infrastructure delivery. Cooperation is a response to sustainable development pluralism. Figure 2 illustrates how the concept of sustainable development in the resource context shapes the procurement of infrastructure projects.

To conclude, PPP procurement is consistent with sustainable development in the context of sustainable human resource management as it means balanced engagement and participation (use, creation, and maintenance) of government and people (individuals and business)—different kinds of human and social resources. These resources can be categorized in terms of the public and private sector. Therefore, we confirm Zhang's et al. findings of the active involvement of both partners as a critical factor for sustainable development of the PPP mode [52] as well as Bjärstig indicating that partnerships improve sustainability [14]. Also, sustainable contractual PPP relationship supports the achievement of social benefits and the improvement of future social values [53].

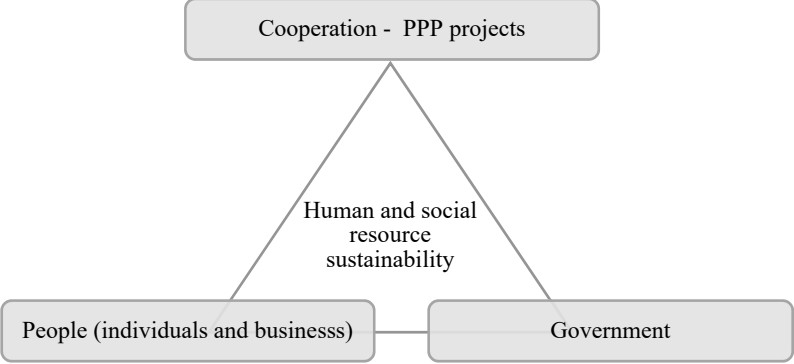

**Figure 2.** Human and social resource sustainability as determinants of Public–Private Partnership (PPP) cooperation (source: own study).

## 3. Public–Private Partnership (PPP) in the Context of the Stakeholder Theory

Public–Private Partnership (PPP) is a way of contracting infrastructure projects. It is usually a long-term relationship between a public sector as procurer and purchaser, and multiple private sector companies which design, construct, and maintain the infrastructure and provide some related services [52,54]. PPP is connected with different arrangements [36], influenced by legal traditions [33,39,40,55]. Generally, the agreements are based on a simple "design-build" (DB) contract for a public utility. The contract may take many forms [33,56], sharing benefits between the partners, proportionately to resources engaged, responsibilities, and risk taken. In this way, public services and infrastructure provision lead to a win–win situation [57]. Figure 3 illustrates typical PPP forms and contracts.

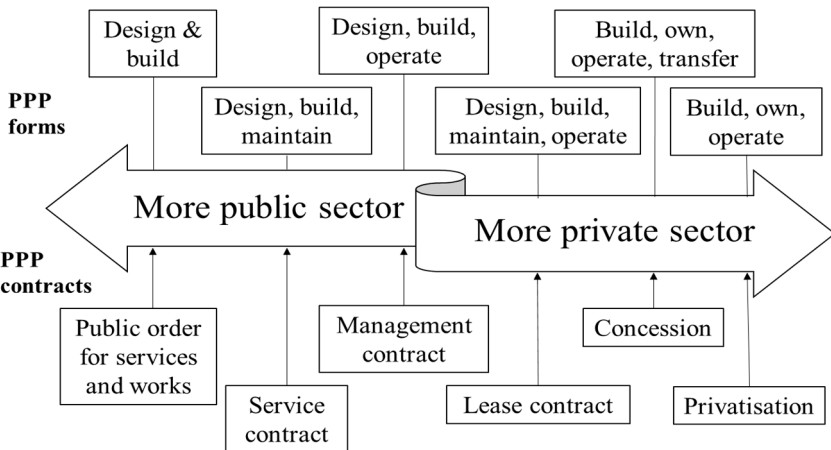

**Figure 3.** PPP forms and contracts (source: own study).

A PPP project is usually developed by a public sector authority (a national or local government) through a bidding (public procurement) process for project agreement. A public sector authority, responsible for infrastructure delivery, acts as a grantor and provides the right to deliver infrastructure services in a PPP form. It also obtains the right to oversee the management of services provision. Simultaneously, public sector authorities are regulators designing the regulatory framework, issuing permits and licenses. Sponsors are equity investors. The public authority (grantor) can be a project sponsor. When more than one sponsor is involved, a joint-venture structure has to be agreed. Typical sponsors include private companies: contractors, suppliers, and operators. Public utility companies (purchasers), delivering the product to final clients, can also support the project as sponsors. Other investors might be investment funds, institutional investors, public agencies, multilateral institutions with different interests related either to a proper rate of return or to stimulation of the local and

regional development. The PPP debt financing is provided mainly by commercial banks and bond investors interested in investment returns. Additional credit support or guarantees might be obtained from public agencies and multilateral development banks [56]. In a typical PPP, project stakeholders include unions (protecting labor resource), media (disseminating and gathering information), and ecologists (protecting environment). The identified parties create dynamic relations in a PPP project. Their participation and engagement may shift over time because of the long-term nature of the PPP project [21]. Figure 4 illustrates typical PPP stakeholders.

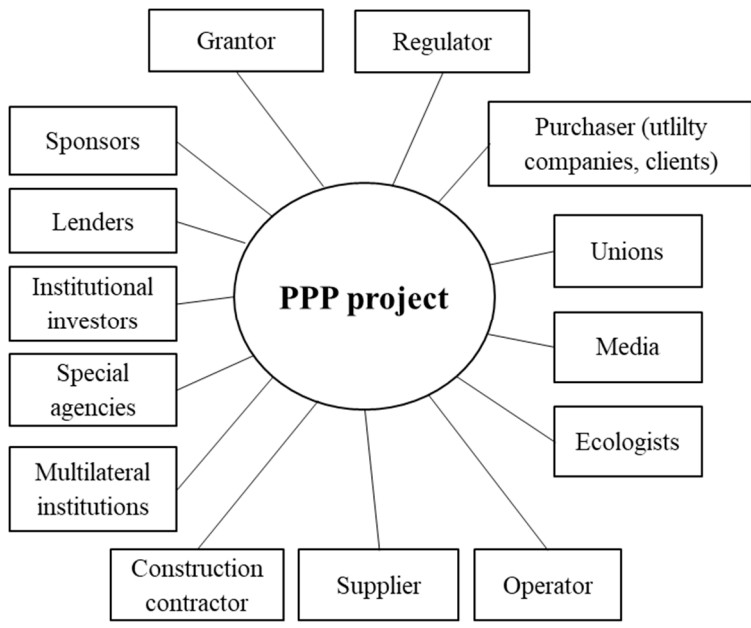

**Figure 4.** PPP stakeholders (source: own study).

The public sector, supervising the project and acting on the behalf of the society, takes care of the total benefits of PPP projects. The benefits should include economic, social, and environmental effects according to sustainability. The private sector is more focused on profits through construction, financing, operation, and other contracted services [58]. PPP is then a coalition of powerful but often conflicting individuals and interest groups. They are PPP stakeholders and they include the entities which have an interest in the project and the ability to influence the project [28,30]. The stakeholders also have a legitimate claim and moral responsibilities [31] and interact with the project thus making its operation possible [59]. According to the classics [29], stakeholders are entities without whose support the organization fails to exist. A strategic approach to a stakeholder concept defined a stakeholder as any individual or group who can affect or is affected by the achievement of the organization's objectives. Finally, the dynamics concept of stakeholders assumes that the mix of stakeholders and their stakes may change over time [32], which mainly applies to a long-term PPP project. Figure 5 presents the types of stakeholders classified according to the attributes of power, urgency, and legitimacy.

Following Mitchell's et al. [32] approach, Schepper et al. [11] developed a model for stakeholder identification for PPP. The model used attributes of power and urgency, and identified three potential types of stakeholders. There are stakeholders who have a minor influence on the project—they do not control critical resources, and their claims do not need immediate attention. There are stakeholders who may have a potential influence on the project—they possess one of the attributes mentioned above. Definitive stakeholders have a direct influence on the project and the environment—they control critical resources and their claims are urgent. Henjewele et al. [12] examined the process of multi-stakeholder consultation and management in a PPP project environment. They highlighted the problem of exclusion of the public sector client in PPP projects. Although Bjärstig [14] claims that cooperation is a promising mechanism for managing stakeholder conflicts, Soomro and Zhang's [21,22]

indicated that conflict of interest is a key PPP failure driver. Schepper, Dooms, and Haezendonck [11] confirmed that stakeholder problems emerge because of the imbalance of reactive and proactive stakeholder management. Guarini and Battisti's claims [10] that development and redevelopment processes based on PPP require a balance between public and collective interests. Stakeholder theory, with its pluralistic nature, can therefore contribute to PPP success by supporting the understanding of PPP stakeholders by their identification and classification. These stakeholders and their attributes will then determine the evaluation of the PPP project, as PPP project success is the sum of the single stakeholder's successes. Concluding, stakeholder theory impacts the evaluation of the PPP project and this evaluation has to consider all stakeholders. Generally, there will be two kinds of inversely related evaluation criteria associated with PPP—"the higher the public benefits; the lower the private surplus" [41] (p. 1). This raises a question of stakeholders' success perception.

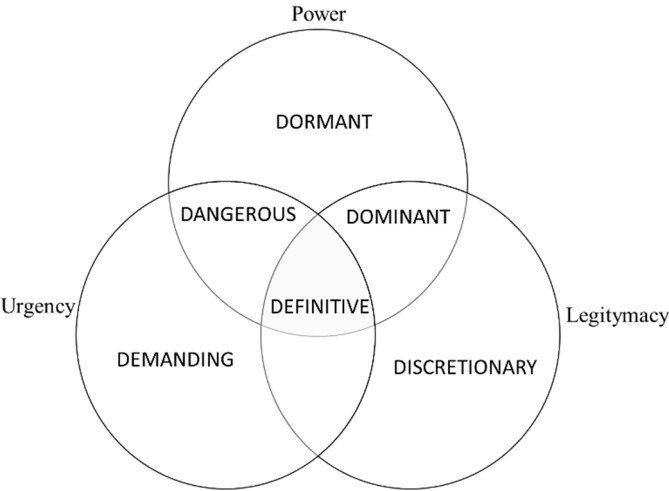

**Figure 5.** Types of stakeholders [32].

## 4. PPP Stakeholders' Success Perception

The success must be related to the purpose [60]. A private sector entity generally strives to maximize the benefits for its owners. The basic evaluation criterion is then the present value of the investment or the rate of return. In the case of projects implemented by public sector entities, the aim of maximizing the benefits is considerably different. The public sector entities function in a kind of dependency [37]. It means that public sector objectives are not self-contained and the public sector entities act in association with the society and "collective interest, of which public administration (PA) is the carrier" [10]. Therefore, conditions of public sector functioning are not just economic and the activities of the public sector should also be assessed on social and natural grounds. As a result, the basic criterion for decision-making is "the public value". It should maximize the benefits for citizens in the interest of which the public sector operates. In such conditions, project success verification is difficult as it goes beyond the functional evaluation of the project itself and public sector decisions must be based on complex criteria. Liang et al. [57] performed a comprehensive review of evaluation criteria for a PPP project and classified them into the following groups: financial strength, technical and management ability, relevant experience, credit level, governmental: insurance, guarantee, credibility, risk-sharing. However, these criteria neglect the input–output relation. A more process-like approach was used by Guarini and Battisti [10]. Claiming that PPP redevelopment initiatives focus more on the satisfaction of the private rather than the public interests, they proposed and tested benchmarking multi-criteria evaluation. Their proposed methodology includes environmental, financial, socio-economic, procedural, and technical (urban) criteria and enables the identification of benchmarks thanks to participation of the different stakeholders mainly through interviews. As "meeting output specification" [19] (p. 5) was acknowledged by the World Bank and Project

Management Institute, this research focuses on success considering the process approach similar to Hueskes et. al. [1], claiming that output specification enables evaluation, monitoring and/or enforcing of performance, and Kumaraswamy's et al. argumentation of output-based payment approaches for publicly funded PPP projects [53].

Three basic success related concepts, relevant to this research and process approach, have been identified. The European Union [34] formulated evaluation criteria for the assessment of structural funds employment. Trocki [38] proposed to adopt this concept and developed it further to success evaluation on three levels: project, organization, and environment. The evaluation criteria include:

- relevance (extent to which project objectives are relevant to identified needs),
- effectiveness (extent to which project outputs are achieved),
- efficiency (relation between resources requires and project output),
- utility (benefits to target groups),
- sustainability (durability of employed effects),
- community added value (the extent to which project outputs, results, and impacts occur due to project intervention).

The success depends on the degree to which these criteria are consistent with expectations, which are obviously different for each stakeholder. The criteria of efficiency and effectiveness, and differentiation between short-term output and long-term outcome (results) are also used in the second concept of project success. Dalcher [35] identified four levels of project success. Level 1 focuses on project management success and uses criteria of profitability and performance measures. Success at Level 2 relates to quality and acceptability of the project output in relation to the stakeholders. Level 3 embraces the creation and delivery of internal value from the business perspective. Level 4 is centerd on prospects relating to future gains and new ventures and opportunities thanks to the project success. It includes the development of new skills, competencies, and capabilities, too. Following the evaluation in the certain scope perspective, Hoge and Greve [36] proposed a conceptual model including five levels of PPP project evaluation: project, delivery, policy, governance, and cultural context. There are objectives related to each level and therefore the success of PPP might be judged at each level. The levels and objectives often overlap and much of the judgment resides outside of the project itself. The project level relates to the objective connected with providing value for money. The delivery refers to the promise of providing goods or services on-time, in-budget, and within the scope. Objectives of infrastructure provision without growing public debt, transfer of risk, application of the more flexible private law, and support of private business relate to the sphere of policy. The governance level means improving accountability and transparency. Finally, cultural context refers, among others, to innovation.

The above concepts confirm that success is perceived quite differently by PPP stakeholders and efficient stakeholder analysis should generate benchmarks useful for measuring the PPP performance and, consequently, the level of satisfaction that stakeholders feel for that same initiative. To conclude, success and satisfaction depend strongly on the scope perspective concerning more internal or external focus. The key criteria dimensions also relate to financial and non-financial benefits, which can be gained in the short and in the long-term. It implicates the identification of the business (financial) and social (non-financial) perspective related to the level of the project (internal focus) and level of the organization and environment (external focus). The interest of each stakeholder will depend on effects availability—there are products and deliverables in a short time, or outcomes/results, and impact in a long time. The variety of approaches raises a question of the possibility of a comprehensive approach to stakeholder analysis in PPP projects.

## 5. A Conceptual Framework of Stakeholder Analysis in PPP Projects—Proposition and Discussion

According to the preceding sections, the complexity of the stakeholder environment increases in the PPP context. According to Varvasovszky and Brugha [61] stakeholder analysis is used to increase

the chances of project success during its preparation, implementation, and during or after project completion for its evaluation. It is applied in the interconnected areas of policymaking, strategic and operational management, and project implementation. "Stakeholder analysis is an approach, a tool or set of tools for generating knowledge about actors—individuals and organizations—so as to understand their behavior, intentions, interrelations, and interests; and for assessing the influence and resources they bring to bear on decision-making or implementation processes" [61] (p. 338). According to Reed et al. [62], the methods used in stakeholder analysis include expert opinions, focus groups, semi-structured interviews, and snow-ball sampling. Their implementation requires proper resources (facilitation, time, setting, training) which also determine their strengths and weaknesses. The methods fall under a top-down or bottom-up approach. However, Miccoli, Finucci, and Murro [15,63] noticed that from recent practice the need for adopting dynamic and non-linear approaches to involve stakeholders, in particular to participation in landscape management and sustainable urban regeneration, has emerged.

Therefore, to understand PPP stakeholders and contribute to PPP success, we develop a conceptual framework of stakeholder analysis. First of all, the framework builds on the sustainable development as the balance of scarce resources claims cooperation contributing to urban sustainable development. Second, the framework builds on the stakeholder theory, taking into consideration the expectations of different stakeholders. We followed the approach presented by Mitchell et al. [32], who identified attributes of power, interest, and legitimacy, to classify stakeholders, and by Schepper et al. [11], who used attributes of power and urgency to identify types of influence. We assumed that there are three key attributes which create frames for stakeholder analysis in PPP. As legitimacy is more static [64], we selected the interest, power, and urgency. Finally, adapting the European Commission [34], Dalcher [35], Trocki [38], and Hoge and Greve [36], our framework takes into consideration the time and scope perspective, that is a short time and a long time perspective, and the level of project, organization and environment. The framework proposes related criteria and an approach to measures and indicators which can be useful for further development of benchmarks measuring PPP performance and stakeholder satisfaction.

The **level of interest** relates to expected PPP's benefits [65]. In this context, both the grantor and the sponsor (strategic equity investor) are characterized by the highest level of interest although expressed differently. Their potential exit from PPP, especially due to their direct engagement in a construction phase, is more complex and very often perceived as a PPP failure [21]. The public sector is also oriented on political success as users are usually voters. While the interest of a single user can be reduced to the access to certain service and goods, the consumers as a group are characterized by a relatively high level of interest. A lower level of interest characterizes other groups of stakeholders, for instance, lenders may sell their debt.

**The power** is connected with the ability to influence the project. It can relate to access to critical resources, such as reputation, competence, or ability to deploy power derived from the position in the organization, or access to funds. There are two PPP stakeholders which can be perceived as "focal" in this context. They are the grantor (local or national government) and a sponsor (private equity investor) [11]. The imbalance between these stakeholders, determined by their power, reflects a potential trade-off between social and market expectations [66]. On the other hand, "win–win" situation, characteristic for PPP projects, may hamper the PPP project as the conflict-free solutions might not be very ambitious, might hinder project progress, and finally delay benefits for all stakeholders [67,68]. Moreover, a general principle is that PPP project results must be consistent with formal requirements related to a certain PPP program in a given jurisdiction. The collective performance of all PPP projects determines whether the PPP program is effective as a strategy or policy for infrastructure development and management. This may also hamper the project [21,22] from the perspective of the private partner as PPP success depends on respecting both sectors' perspectives.

**The component of urgency** helps to move the framework from static to dynamic. The urgency is the degree to which stakeholder claims call for immediate attention [32]. It is based on time sensitivity

(the degree to which managerial delay in attending to the claim or relationship is unacceptable to the stakeholder) and criticality (the importance of the claim or the relationship to the stakeholder). In PPP the responsibility related to urgency is shared between public and private partners by different forms of PPP contracts. Assuming that urgency is a claim for immediate attention and therefore can be expressed by the ability to control the environment, we aggregate urgency and power into engagement.

By combining engagement and interest with time (short-term, long-term) and scope perspective (internal or external focus), we identified four different types of potential homogeneous stakeholders and the appropriate success criteria relating to project management, business/organization, environment, and future. Figure 6 presents the conceptual framework.

| | | | Preferred benefits (interest) | |
|---|---|---|---|---|
| | | | **Financial** (short-term benefits - products/deliverables) | **Non-financial** (long-term benefits—outcomes/results, impacts) |
| **Engagement** | **Direct** (internal focus) | | **Type I** (strategic sponsor, construction company, sponsor) <u>Project management success</u> (internal measures for efficiency and performance criteria related to budget, schedule, specification) | **Type II** (grantor) <u>Business/organization success</u> (creation of value for citizens due to delivered products and services, relevance, utility and effectiveness, criteria) |
| | **Indirect** (external focus) | | **Type III** (investment funds, institutional investors, public agencies, multilateral institutions, subcontractors, operators, suppliers, off-takers, advisors) <u>Environment (external entities) success</u> (quality and acceptability of benefits for stakeholders, efficiency) | **Type IV** (media, ecologists, unions, governmental supervisors and regulators) <u>Future potential success</u> (future opportunities, wider implications for stakeholders, socio-economic impact, durability criterion) |

**Figure 6.** The conceptual framework for stakeholder analysis in PPP projects (source: own study).

Based on the engagement, we identified two types of stakeholders—direct (with more internal focus) and indirect (with more external focus). Based on the different perception of interest, the research assumes that PPP can bring two types of preferred benefits which differ in relation to timing. There are more financial and short-term focused stakeholders and more non-financial and long-term focused stakeholders. This allowed us to distinguish four types of stakeholders.

Stakeholders who are grouped as Type I and Type II are the most engaged in the PPP project. There are two key stakeholders who are Type I (the strategic sponsors and construction company, often being the sponsor) and Type II (grantor). Their engagement (power and urgency) related to the PPP is the most prominent. When their support is withdrawn, the project usually fails. All of them (Type I and II) are interested in the financial and technical feasibility of the project, which is a condition for an expectation of long-term benefits of a more non-financial nature. Efficiency and performance metrics measure the financial and technical feasibility of the project. However, for the grantor, who is ultimately responsible for service delivery, non-financial benefits are relevant, too. The criteria of evaluation refer then to relevance, utility, and effectiveness, representing value for citizens.

Remaining investors (investment funds, institutional investors, public agencies, multilateral institutions), subcontractors, operators, suppliers, off-taker (utility companies), advisors are Type III. Their participation may change during the operation phase which does not have to lead to PPP termination. Resigning from services by a single user (a utility company's client) probably would not cause substantial harm to the project. However, a group of consumers may have enough power to claim their needs (and therefore can change their status to Type II). Lenders and bond investors are grouped in Type III, although their particular influence is determined by the scale of financial

engagement. The bigger the financing leverage, the greater the engagement, which changes their status to Type I. Type III of stakeholders evaluates the project basically in terms of the efficiency. Type IV covers media, ecologists, and unions influencing the success perception in the widest context. They are main externally focused stakeholders without a direct financial interest in the project. Type IV also includes governmental supervisors and regulators. They evaluate the effectiveness, durability, and the socio-economic impact of the PPP.

The conceptual framework proposes an approach to benchmark satisfaction within four identified categories of homogeneous stakeholders. Project management success is basically measured by internal efficiency which refers to the relation between resources requirements and project output. The resources relate to budget, schedule, and project specification. They include financial, human, and material input mobilized for the implementation of the project [41]. The input data is processed with a thorough analysis, in particular a financial analysis, to develop mainly the quantitative performance measures [50,69,70]. Project management success can be also evaluated based on the approach developed by the Project Management Institute [42] which can be applied to PPP projects [19].

Business/organization success refers to the creation of value for citizens due to delivered products and services. The criteria, which include relevance, utility and effectiveness, refer to the relations between identified needs and project objectives (relevance), identified needs and effects for the target groups (utility), and project objectives and effects (effectiveness). The input data require then collecting information about needs, problems and issues identified by the society, objectives of the PPP project, and the long term effects which are the project outcomes/results and impacts. The data should be processed with thorough analysis, including multicriteria analysis [10,41], to develop indicators of whole business/organization performance going beyond the project itself, but still of internal character. These indicators are important to an organization in achieving its strategic goals, objectives, vision, and values [70], namely value for citizens.

The evaluation of the environment (external entities) success uses criteria of quality and acceptability of benefits for stakeholders, and also efficiency. However, these criteria are more external-like. They require input data, such as financial, human, and material means, but with the concern of economic externalities related to the project. Multicriteria analysis would be appropriate in this case too.

Finally, future potential success related to future opportunities means broad implications for stakeholders and new initiatives inspired by the PPP project. The success here is evaluated based on the socio-economic impact and durability criterion. Durability is similar to the utility as both criteria refer to the relation between needs and effects but durability is more externally focused. Socio-economic impact means including social and environmental externalities which are the input data. They are important, particularly for media, unions, and ecologists. Organizational and regulatory means are a sort of input delivered by governmental supervisors and regulators.

To conclude, input data, which must be collected from the stakeholders analysis, include financial, human, material, organizational and regulatory resources, externalities of economic, social and environmental character, information about project objectives and effects, and last but not least, the information about needs/problem/issues, which should be the generic reason for the PPP project development. The input data collected in the stakeholder analysis should be then processed with the proper method and techniques. However, as the nature of PPP development is often difficult to describe in standard categories, quantitative evaluations which aim to provide simple benchmarking measures are limited. Methods reflecting the stakeholders' different points of view, exemplified—but not limited to multicriteria analysis—can be recommended. Finally, the framework output data represent together shared, accepted, and sustainable PPP because the benefits are dependent and available for different types of stakeholders. Moreover, the relation between the benefits has a forward and cyclical character. It means that the short-time benefits, which are represented by project products and deliverables, are essential to gain long-term benefits. They are represented by project outcomes/results and finally, impacts. This process of cumulative benefits has a cyclical character, as project impacts, referring to future opportunities and emerging expectation, are about stimulation of new PPP initiatives, which,

closing the cycle, generate products and deliverables in the short-time, and outcomes/results in the long-time, which impact the future.

## 6. Conclusions and Limitations

Given the space for theoretical discussion, the conclusions link back into that process. This article was based on sustainable development concept and stakeholder theory, which indicate that there is more than one approach to understand and implement sustainability, and there is more than one approach related to project stakeholders analysis. These issues are important in the context of PPP projects, implementing sustainable development principles (sustainable use, maintenance, and creation of resources) and also contributing to sustainable development (delivering effects corresponding to different stakeholders' expectations according to triple bottom). During PPP project preparation, realization, and operation, we witness the process of creation of new human and social resources. New competencies, knowledge, and networks develop and these can be applied for future benefits. However, human resources and social capital, determining project success, also require balance in terms of interest, power, and urgency. Conducted a literature review reveals that the problem of joint cooperation between the public and private sector under the condition of sustainable development requires stakeholder centered analysis. Striving to build a new approach to stakeholder analysis in PPP projects as a condition of sustainable urban development, the research tried to provide an answer for three research questions: How does the sustainable development shape the procurement of infrastructure projects?, How does the stakeholder theory impact the evaluation of PPP infrastructure projects?, How do PPP stakeholders perceive the project success? Simultaneously, the limitations were identified.

First, we claim that sustainable development shapes the procurement of infrastructure projects towards PPP, rather than a conventional form of sole public sector infrastructure delivery. PPP is a response to sustainable development pluralism as government and people (individuals and business), participating and engaged in PPP, are valuable human and social resources. They are a component of scarce and rare resources determining, and simultaneously requiring, sustainable development. They become capital in managing sustainable development through sustainable infrastructure projects on the condition of their balanced use, management, and creation. This can be gained by PPP cooperation, engaging public and private sectors with respect to their diverse needs. Sustainable development is also determined by acting "smart". Thus human and social resources, which are components of sustainable development, determine the cooperation between the city authorities and other entities, shaping the procurement of infrastructure projects. The limitation of the research in this extent is lack of a discussion concerning the concept of "balanced", as "balanced" does not mean "equal". The balance in the context of infrastructure delivery depends on the type of infrastructure and requires a proper form of contract as illustrated in Figure 3, where risk and benefits are balanced, but not necessarily shared equally. This is the subject of another research.

Secondly, we state that the stakeholder theory impacts the evaluation of PPP projects. Local governments, community-based organizations, foundations, neighborhood and other advocacy groups, construction companies, investors, commercial banks, tenants and their brokers, ecologists, media and unions, all are the participants of the city development and city stakeholders. They should be able to create feasible projects which generate benefits and reduce the risk involved in urban development. Urban development policy is a result of decisions made by specialists, including public authorities and professional investors. However, attention must be paid to a wide range of social groups. The stakeholder theory emerged in this context. Inhabitants and infrastructure users are no longer simply clients or even voters. They are also co-decision-makers, contributing to the development of infrastructure projects. Stakeholder theory, with its pluralistic nature, supports PPP stakeholder analysis and therefore determines the PPP evaluation as PPP project success is a sum of single stakeholder's successes. This means that according to the stakeholder theory, the evaluation of the PPP project has to consider all stakeholders. The limitation of the research in this extent is that as

stakeholder theory has an implicit pluralism present at its heart, it has been chosen as the basis on which the social relations and the subject under study can be understood. However, that is not the only possible approach, for instance, a political economy could also frame the study of Public–Private Partnership [54,71].

Third, we proved that stakeholders groups perceive PPP success differently. The finding identifies apparent discrepancies in the perception of PPP success between main PPP stakeholder groups. The essence of PPP is a combination of private capital, private project execution and the delivery of public services and/or goods. A PPP is a combination of the tasks and objectives of the two sectors: public and private (more social-focused and more commercial-focused). The public sector invests within the PPP because of social well-being, while the private sector partners expect rates of return. Thus, PPP investments must represent a beam of aims and can be successful if all stakeholders are successful. The limitation of the research to this extent is that the research does not describe situations in the middle: whether the PPP project, delivered in time and scope, but not in the budget, will be successful or not; whether the PPP project, which is relevant and efficient, but not effective, will be successful or not. At this stage of research, the successful PPP project must meet all criteria which in practice might be difficult. Therefore, further research is needed in that respect.

Finally, building on a concept of sustainable development and stakeholder theory, we propose a conceptual framework to stakeholder analysis in PPP projects as a condition of sustainable urban development. Our approach combines two main dimensions which are stakeholder engagement (power and urgency) and perception of interest, with time and scope perspective, and distinguishes four types of potential homogeneous stakeholders and the appropriate success criteria relating to project management, business/organization, environment, and future. The limitation of the framework might be a fact that in a case of multiple stakeholder perspectives, an effective "partnership" composition should be truly "external" and the scope perspective of "internal" and "external" foci might not be appropriate. Therefore, the framework is rather "evolutionary", starting in the top left and extending as the process becomes more sophisticated. Lastly, the operationalization of the framework needs benchmarks for each single judgment criteria with which a PPP is evaluated. Further research, which is needed in that respect, is already in progress.

As a final point, we assume that the framework can contribute to a better understanding of determinants of sustainable city development and PPP success. It proves its value added as it organizes a number of relevant contents, however it is just a proposition in the discussion relating to improvement of comprehensive city management. It could help public and private partners to understand that success, built on trust (social capital), eventually leads to win–win projects over sectorial boundaries when built according to sustainable development and stakeholder theory. However, new questions arise. They concern how sustainable development principles apply to different PPP projects, what is the other basis on which PPP can be discussed so that more recommendation can be made, and finally, how to improve PPP evaluation. The study proposes a background for further research on PPP success and crucial stakeholders. In the end, we need to say that the concept needs operationalization and empirical validation. That requires observations and in-depth interviews related to different types of PPP case studies. The future research could also consider characteristics of specific countries circumstances as the need to adopt more stakeholder-oriented perspective resulting from many participants involved, but also from different PPP form, contracts and legal tradition.

**Author Contributions:** A.W.-F. 55% and J.W. 45%: Conceptualization, A.W.-F. and J.W.; Formal analysis, A.W.-F. and J.W.; Funding acquisition, J.W.; Investigation, A.W.-F. and J.W.; Methodology, A.W.-F. and J.W.; Project administration, A.W.-F. and J.W.; Resources, A.W.-F. and J.W.; Supervision, A.W.-F. and J.W.; Visualization, A.W.-F. and J.W.; Writing—original draft, A.W.-F. and J.W.; Writing—review & editing, A.W.-F. and J.W.

**Funding:** This research was funded by the Faculty of Management of the University of Gdańsk, and Faculty of Economics and International Relations of the Cracow University of Economics.

**Conflicts of Interest:** The authors declare no conflict of interest.

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
