# Peer review of "Understanding of Public–Private Partnership Stakeholders as a Condition of Sustainable Development"

_sustainability, doi:10.3390/su11041194_

Round 1
Reviewer 1 Report
I read the paper that deals with interesting topics that are subject of my research in recent years; this topic are related, in summary, to the inclusion of stakeholders in the PPP processes, to increase the level of success and sustainability of the PPP initiatives themselves.
The article, however, can not be published until after a major revision because it is lacking essential elements to be able to be configured as an advancement in the field of research in this topic.
Considering that the first 4 parts of the article are substantially a summary of bibliographic research, the progression in the research that this work should be reach is represented by the "conceptual model" which, however - as presented - can not be considered an operating model but only a series of basic informations and assumption, taken from surveys bibliographies, for the future construction of a model that can be implemented and tried out.
A model aimed at implementing an efficient stakeholder analysis must “produce” benchmarks useful for measuring the performance of a PPP initiative and, consequently, the level of satisfaction that stakeholders feel for that same initiative.
Therefore, to be able to configure itself as an advancement in the field of research, the "conceptual model" should be operationally developed deepening the following concepts:
- what is the final objective of a stakeholder analysis: for example, obtain output data (benchmarks) and how to use them; I think it is important, for example, that the output data of a stakeholder analysis - to be meaningful - must represent the performances that can generate a certain level of satisfaction for the various stakeholders (organized in categories). The benchmarks can therefore be obtained for each single judgment criteria with which a PPP is evaluated;
- how to implement a stakeholder analysis (questionnaires, meetings, interviews, public assemblies, etc.); in this regard we can see the works of authors such as Bobbio Luigi, Miccoli Saverio, Mattia Sergio, Oppio Alessandra, and Guarini Maria Rosaria.
- you correctly talk about "categorization" of the stakeholders; it would be interesting to propose a list of "categories" of homogeneous stakeholders that can be included in the PPP processes;
- what are the input data that must be collected among the stakeholders in a stakeholder analysis;
- how to process input data collected in the stakeholders analysis to obtain output data that represent, as already stated, useful elements for assessing the level of satisfaction and sustainability (in respect of the stakeholders' point of view) of a PPP;
- how output data can alternatively represent basic elements for the construction of a shared, accepted and sustainable PPP.
Best Regards
Author Response
Dear Reviewer,
We would like to thank for the valuable comments and suggestions which encouraged us to discuss, deepen and clarify important aspects of the research. In particular, the remarks concerning the idea of the “conceptual model” as a series of information for future model construction and its future implementation, encouraged us to present the value-added of the concept more clearly. Secondly, the research could be improved thanks to the inclusion of the recommended bibliography which not only gave more light on the research and let us deepen its justification but also convinced us that the research is needed. The revised text has been also reviewed in terms of the language and style. We provide detailed answers in the text of the review below, starting with annotation “Authors’ answer”.
Best regards,
authors

Reviewer 2 Report
Initially I appreciate the editor's consideration for the revision of this interesting work entitled: "Understanding of Public-Private Partnership Stakeholder as condition of Sustainable Development". Certainly, this paper is an important contribution that analyzes the hypothesis of the significant achievement of a sustainable development in the city through the understanding of the PPP Stakeholders. The paper develops a fundamentally theoretical analysis to answer three linked research questions. The structure and writing of the paper is adequate and allows the reader to follow the purpose of the paper. So also for the results achieved are honest and the limitations derived from the need for an empirical study are defined. However, in the opinion of this reviewer, bellow some concerns that allow improving this contribution are exposed.
-Some references are identified with more than 10 years old. It would be appropriate to use more up-to-date literature in consideration of the broad contribution to knowledge in the topics of Sustainable Development and Theory of Stakeholders in recent years.
-In relation to the above, the theoretical analysis is based on certain proposed authors that condition the result of this paper. Accordingly, it would be appropriate to substantiate specifically because these authors are adequate and not others. Clarifying this point would improve the reliability of the analysis achieved.
Author Response
Dear Reviewer,
We would like to thank for the valuable comments and suggestions which encouraged us to discuss, deepen and clarify important aspects of the research, in particular, the remarks concerning the literature. We provide detailed answers in the text of the review below, starting with annotation “Authors’ answer”.
Best regards,
authors

Round 2
Reviewer 1 Report
I think that this new version of the article solves the lacks highlighted in the first version. I still think it would be very interesting to propose a model but probably the article could be too lenghty; this could be the subject of further study and further work. The article can be accepted. Best Regards